# Evaluation of arsenic metabolism and tight junction injury after exposure to arsenite and monomethylarsonous acid using a rat *in vitro* blood–Brain barrier model

**Hiroshi Yamauchi***, **Toshiaki Hitomi, Ayako Takata**

Department of Preventive Medicine, St. Marianna University School of Medicine, Kawasaki, Japan

* hyama@marianna-u.ac.jp

**Data Availability Statement:** All relevant data are within the paper and its Supporting Information files.

## Abstract

Experimental verification of impairment to cognitive abilities and cognitive dysfunction resulting from inorganic arsenic (iAs) exposure in children and adults is challenging. This study aimed to elucidate the effects of arsenite (iAs$^{III}$; 1, 10 and 20 μM) or monomethylarsonous acid (MMA$^{III}$; 0.1, 1 and 2 μM) exposure on arsenic metabolism and tight junction (TJ) function in the blood–brain barrier (BBB) using a rat *in vitro*-BBB model. The results showed that a small percentage (~15%) of iAs$^{III}$ was oxidized or methylated within the BBB, suggesting the persistence of toxicity as iAs$^{III}$. Approximately 65% of MMA$^{III}$ was converted to low-toxicity monomethylarsonic acid and dimethylarsenic acid via oxidation and methylation. Therefore, it is estimated that MMA$^{III}$ causes TJ injury to the BBB at approximately 35% of the unconverted level. TJ injury of BBB after iAs$^{III}$ or MMA$^{III}$ exposure could be significantly assessed from decreased expression of claudin-5 and decreased transepithelial electrical resistance values. TJ injury in BBB was found to be significantly affected by MMA$^{III}$ than iAs$^{III}$. Relatedly, the penetration rate in the BBB by 24 h of exposure was higher for MMA$^{III}$ (53.1% ± 2.72%) than for iAs$^{III}$ (43.3% ± 0.71%) (p < 0.01). Exposure to iAs$^{III}$ or MMA$^{III}$ induced an antioxidant stress response, with concentration-dependent increases in the expression of nuclear factor-erythroid 2-related factor 2 in astrocytes and heme oxygenase-1 in a group of vascular endothelial cells and pericytes, respectively. This study found that TJ injury at the BBB is closely related to the chemical form and species of arsenic; we believe that elucidation of methylation in the brain is essential to verify the impairment of cognitive abilities and cognitive dysfunction caused by iAs exposure.

## Introduction

Chronic arsenic poisoning, caused by contamination of drinking water with inorganic arsenic (iAs), has occurred on a large scale in Asian and Latin American countries [1]. Epidemiological studies in Bangladesh [2,3], Taiwan [4], and Mexico [5] have identified the impairment of cognitive abilities of children as an emerging health issue in areas where chronic arsenic

**Funding:** This work was supported by JSPS KAKENHI Grant No. JP21H03185. Ayako Takata." And we report Takata's role. Takata had the roles of Data Curation, Formal Analysis, Investigation, Supervision, Validation, Visualization, and Writing-Review & Editing.

**Competing interests:** The authors declare no conflicts of interest.

poisoning occurs. The WHO has also warned about the effects on cognitive development caused by iAs exposure during the fetal stage and infancy [6]. Furthermore, in non-iAs contaminated areas such as Spain [7] and the USA [8], the effects of dietary iAs on children's cognitive abilities have also been studied. In this context, the European Food Safety Authority is concerned about the impact of even low concentrations of iAs in rice and processed rice products on the cognitive abilities of young children [9]. On the other hand, cognitive dysfunction in adults in areas known to be affected by chronic arsenic poisoning has also been reported [10–12]. Furthermore, a follow-up study of 23,000 subacute arsenic-poisoning patients who consumed iAs-contaminated powdered milk for three consecutive months while their blood–brain barrier (BBB) was immaturely revealed cognitive dysfunction that manifested when they were older [13]. These studies highlight that cognitive dysfunction from iAs exposure is a problem for children and adults.

Several technical problems exist when conducting experimental validation of findings from epidemiological studies. Animal studies of iAs exposure and central nervous system damage have confirmed that orally administered arsenite (iAs$^{III}$) can penetrate the BBB via the blood and migrate to the brain [14,15]. In addition, iAs$^{III}$ has been reported to cause tight junction (TJ) injury [16] and, subsequently, neurons [17] as it penetrates the BBB. Detailed information on the permeability and TJ injury at the BBB is essential for elucidating how cognitive dysfunction is induced by iAs exposure. However, there are technical difficulties in measuring the quantitative permeability of iAs at the BBB in animal experiments and in assessing TJ injury. An *in vitro*-BBB model, developed from the triplicate culture of rat [18,19], pig [20,21], and mouse [22] primary brain cells (i.e., vascular endothelial cells, pericytes, and astrocytes), is a promising method for such validation studies. Recent studies have reported the application and effectiveness of the *in vitro*-BBB model, which enables the prediction of the permeability of different substances at the BBB [23,24]. Previous research results using rat *in vitro*-BBB models have included silver nanoparticles [25] and dioxin [26]. However, to date, no research on arsenic exposure and TJ injury of BBB have yet used this rat *in vitro*-BBB model.

It is hypothesized that the health consequences for patients with acute [27] and chronic arsenic poisoning depend on arsenic methylation capacity, which is the efficiency with which iAs are converted to dimethylarsenic acid (DMA$^V$) [28–30]. iAs are methylated by a methyltransferase (AS3MT) and glutathione to produce monomethylarsonous acid (MMA$^{III}$) as the first intermediate metabolite [31,32]. Importantly, MMA$^{III}$ is considered more toxic than iAs$^{III}$ [31–33]. On the other hand, MMA$^{III}$ has been reported to be metabolized in human brain cells, resulting in its conversion to DMA$^V$ [33].

We hypothesize that the impairment of cognitive abilities and cognitive dysfunction caused by iAs exposure occurs in response to TJ injury at the BBB. This is followed by penetration of iAs and its metabolites through the BBB, ultimately leading to glial cell and neuron injury. This study aimed to elucidate the mechanism of TJ injury at the BBB due to iAs$^{III}$ or MMA$^{III}$ treatment using a rat *in vitro*-BBB model.

## Materials and methods

### Chemicals

iAs$^{III}$ (arsenite; arsenic trioxide, 99.995% purity, CAS No. 1327-53-3) was purchased from Sigma-Aldrich (St. Louis, MO, USA). MMA$^{III}$ (CH$_5$AsO$_2$, ≥ 95% purity, CAS No. 25400-23-1) was purchased from Toronto Research Chemicals (Toronto, Canada). Arsenic stock solutions were prepared using cell culture-grade sterile water (Nacalai Tesque, Inc., Kyoto, Japan) and brain capillary endothelial cell culture medium (PharmaCo-Cell Company Ltd, Nagasaki, Japan). Diluted solutions were used for each experiment.

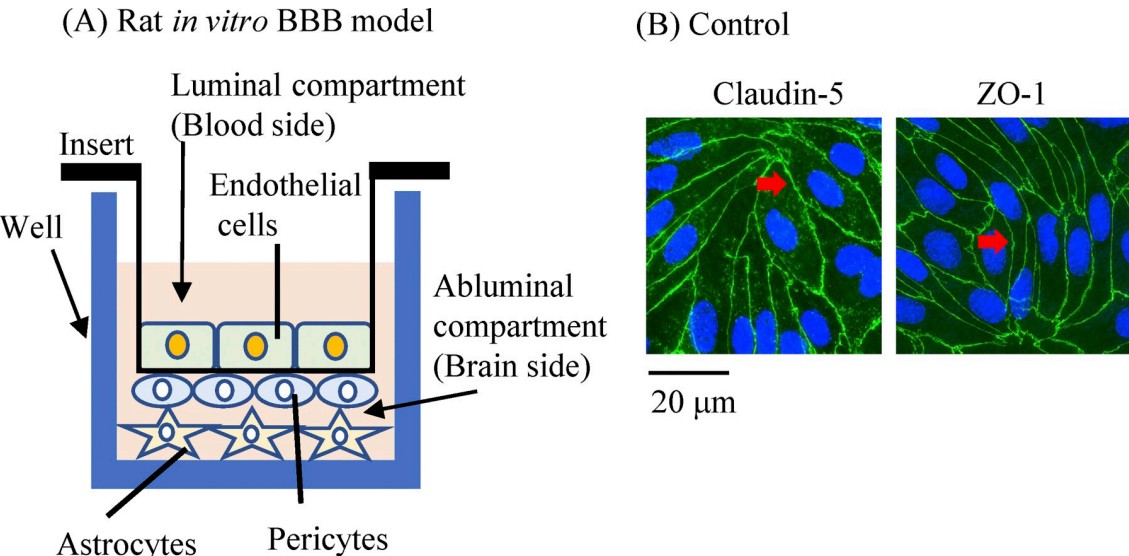

**Fig 1. Structure of the rat *in vitro* BBB model and experimental setup.** (A) Schematic representation of the rat *in vitro* BBB model (luminal compartment; blood side, abluminal compartment; brain side). (B) BBB formation was confirmed by measured transendothelial electrical resistance (TEER) values and by immunofluorescence staining the expression of TJ proteins [claudin-5 and zonula occludens-1 (ZO-1); green, Alexa Fluor 488].

### Rat *in vitro*-BBB model study

A rat *in vitro*-BBB model comprising three cell types (RBT-24H) was purchased from PharmaCo-Cell Company Ltd [18,19]. This rat *in vitro*-BBB model is produced by a triple co-culture of primary brain capillary endothelial cells, pericytes, and astrocytes from Wistar rats. In general, rats are not the best model for arsenic clearance studies because of the strong binding properties of rat erythrocytes to arsenic [34,35]. In the present study, we used primary cultured rat brain cells washed with intracerebrovascular blood to ensure that rat erythrocytes do not bind arsenic and affect clearance. The structure of the rat *in vitro*-BBB model consists of brain capillary endothelial cells on the top of a transwell filter in the insert portion, pericytes on the bottom, and astrocytes attached to the bottom of a 24-well plate (Fig 1). BBB maturation status was then determined by measuring transendothelial electrical resistance (TEER) (Millicell ERS-2, Merck Millipore, Billerica, MA, USA). To do so, cells were first incubated at 37˚C according to the manufacturer's protocol until the TEER value was $150\,\Omega \times cm^2$ or higher. iAs$^{III}$ (1, 10, and 20 μM) or MMA$^{III}$ (0.1, 1, and 2 μM) were exposed in the luminal compartment (i.e., the blood side), and samples taken from the medium in the abluminal compartment (i.e., the brain side) at 6 h and 24 h time points. Samples for Western blot (WB) analysis were collected separately from vascular endothelial cells and pericyte group, or astrocytes. Cells and arsenic-containing solutions were then frozen in liquid nitrogen and stored at −80˚C until further analysis.

### Measurement of BBB intercellular transport

We measured the permeability coefficient of sodium fluorescein (Na-F), a measure of paracellular transport in the rat *in vitro*-BBB models at 24 h after iAs$^{III}$ or MMA$^{III}$ treatment. All measurements were performed according to the manufacturer's protocol [18,19]. Na-F (10 μg/mL) solution was added to the luminal compartment of the insert, and this insert was then transferred to another 24-well plate containing DPBS-H [10 mM HEPES, 25 mM glucose in

Dulbecco's phosphate-buffered saline (DPBS)]. After incubation at 37°C for 30 min, the concentration of Na-F was measured using a microplate luminometer ARVOx4 2030 Multilabel Reader (Perkin Elmer, Waltham, MA, USA; excitation wavelength: 485 nm, emission wavelength: 535 nm). The permeability coefficient of Na-F was calculated using the following equation:

$$P_{app}(cm/s) = V_A/A \text{ x } [C]_{Luminal} \text{ x } \Delta[C]_{Abluminal}/\Delta t$$

Here: $P_{app}$, permeability coefficient. $V_A$, volume of the abluminal chamber; A, membrane surface area; $[C]_{Luminal}$, initial luminal tracer concentration; $[C]_{Abluminal}$, abluminal tracer concentration; t, time of experiment.

## Western blot analysis

For WB analysis, frozen transmembranes containing vascular endothelial cells and pericytes group, and astrocytes were lysed in CelLytic M (Sigma-Aldrich) containing a protease inhibitor cocktail (Nacalai Tesque, Inc.) to extract total protein. After obtaining total protein samples, their concentrations were determined using a CBB solution-based protein assay (Nacalai Tesque, Inc.). Each protein lysate (10 μg/lane) was loaded onto a NuPAGE Novex 3%–8% Tris-acetate gel (Life Technologies Corporation, Carlsbad, CA, USA) and electrophoresed. Protein bands were then transferred to a polyvinylidene fluoride (PVDF) membrane (Invitrogen, Carlsbad, CA, USA) and incubated in Bullet Blocking One (Nacalai Tesque, Inc.) for five minutes at room temperature to block nonspecific binding sites. Membranes were then incubated with the following primary antibodies: rabbit monoclonal anti-nuclear factor-erythroid 2-related factor 2 (Nrf2) (1:1,000, #33649; Cell Signaling Technology, Inc., Danvers, MA, USA), rabbit polyclonal anti-heme oxygenase-1 (HO-1) (1:1,000, 10701-1-AP; Proteintech Group, Inc., Chicago, IL, USA), mouse monoclonal anti- claudin-5 (1:1,000, 35–2500; Thermo Fisher Scientific, Rockford, IL, USA), rabbit polyclonal anti-ZO-1 (1:500, 61–7300; Thermo Fisher Scientific) and mouse monoclonal anti-β-actin (1:2,500, A5316; Sigma-Aldrich). All incubations were performed at 4°C and incubated with secondary antibodies (1:5,000, NA9311ML, or NA93401ML; GE Healthcare, Little Chalfont, UK) for 1 h at room temperature. Positive protein bands were enhanced using a chemiluminescence kit ECL Select Western Blotting Detection Reagent (GE Healthcare) or an ECL Prime Western Blotting Detection Reagent (GE Healthcare). Images were then acquired using an Image Quant LAS 4000 (GE Healthcare). Finally, band density quantification was performed using NIH ImageJ (National Institutes of Health, Bethesda, MD, USA).

## Immunocytochemistry

We used immunocytochemistry to determine the presence of claudin-5 and ZO-1. For claudin-5, vascular endothelial cells on transwell inserts were first fixed with a 50% acetone/50% methanol (1:1, ice-cold) solution for one minute at room temperature, and for ZO-1, cells were fixed in 3% paraformaldehyde in DPBS (+/+), DPBS with calcium and magnesium, for 10 min at room temperature then treated with 0.1% Triton X-100 in DPBS (+/+) for 10 min at room temperature. Both cell samples were then washed in DPBS (+/+) and blocked overnight at 4°C with 3% BSA in DPBS (+/+). Primary antibodies [mouse monoclonal anti-claudin-5 (1:100, 35–2500; Thermo Fisher Scientific) and mouse monoclonal anti-ZO-1 (1:100, 33–9100; Thermo Fisher Scientific)] were diluted in 0.1% BSA in DPBS then incubated at 37°C for 45 min. After washing, samples were incubated with a secondary antibody (1:200, A28175; Goat-anti-mouse IgG H&L, Alexa Fluor 488, Thermo Fisher Scientific) at 37°C for 45 min. DAPI nuclear staining was performed after washing. Finally, all specimens were imaged using a

KEYENCE BZ-X710 fluorescence microscope (Keyence, Inc., Osaka, Japan). Immunofluorescence staining quantification was performed using NIH ImageJ.

## Analysis of arsenic by HPLC-ICP-MS

All exposed arsenic samples were collected from the abluminal compartment (brain side) of the 24-well plate (Fig 1), transferred to a Falcon tube, and the volume was increased to 10 mL by adding Milli-Q water. Samples were filtered through 0.45 μm membrane filters (Hillex-HP, PES 33 mm, Non-sterile, Millipore, MA, USA) before analysis. Next, arsenic species in the filtered medium were separated by high-performance liquid chromatography (HPLC, Agilent 1260, Agilent Technologies, Inc., Tokyo, Japan). Separation by HPLC was carried out using a CAPCELL PAK C18 MG column (4.6 × 250 mm, OSAKA SODA, Osaka, Japan) and a mobile phase solvent (10 mmol/L sodium 1-butane sulfonate, 4 mmol/L tetramethylammonium hydroxide, 4 mmol/L malonic acid, 0.05% methanol, adjusted to pH 3.0). For HPLC, the column temperature was maintained at room temperature, the sample injection volume was fixed at 20 μL, and we used a flow rate of 0.75 mL/min. Arsenic was then detected via inductively coupled plasma mass spectrometry (ICP-MS, Agilent 8800, Agilent Technologies, Inc.).

## Statistical analyses

Statistical analyses were performed using IBM SPSS Version 25.0 (IBM Corp, Armonk, NY, USA). All results are summarized as mean ± standard deviation (SD). Student's t-tests were used to assess the statistical significance of differences in mean values of groups, and one-way ANOVAs with Tukey's or Dunnett's post hoc tests were used to compare two or more groups. Spearman's rank correlation coefficient was used to calculate the association between pairs of variables. Finally, a p-value of $<0.05$ was considered statistically significant for the two-tailed test.

# Results

## Metabolism and penetration rate of arsenic

The blood side (i.e., the luminal compartment) of the rat *in vitro*-BBB model was incubated with 20 μM (3,960 ng/mL) $iAs^{III}$ or 2 μM (248 ng/mL) $MMA^{III}$, and arsenic that crossed the BBB after 6 h or 24 h was then collected from the brain side (i.e., the abluminal compartment). Arsenic content was then quantified using HPLC-ICP-MS in different chemical forms. Penetration rate values at the BBB following $iAs^{III}$ or $MMA^{III}$ treatment are shown in Table 1. The total penetration rate of $MMA^{III}$ (23.1% ± 7.76%) at 6 h was approximately twice that of $iAs^{III}$ (12.9% ± 2.23%), but this difference was not statistically significant. In contrast, a trend toward increased total penetration rate was observed for $MMA^{III}$ (53.1% ± 2.72%) compared to $iAs^{III}$ (43.3% ± 0.71%) in the 24 h samples ($p < 0.01$). Next, we summarize the metabolism of $iAs^{III}$ or $MMA^{III}$ shown in Table 1. The final metabolite of $iAs^{III}$ or $MMA^{III}$ was $DMA^V$, but no dimethylarsinous acid was detected. The unconverted rate of $iAs^{III}$ after 24 h was 84.6%, a characteristic of the low rate of metabolism observed. In contrast, for $MMA^{III}$ the unconverted rates at 6 h and 24 h were 34.6% and 25.9%, respectively. These results for $MMA^{III}$ suggested a feature that makes it more readily susceptible to methylation and oxidation reactions than $iAs^{III}$. Furthermore, these results suggest important information that $MMA^{III}$ was not degraded to iAs in the BBB. In this study, $iAs^{III}$ or $MMA^{III}$ were oxidized and converted to arsenate ($iAs^V$) or monomethylarsonic acid ($MMA^V$), but possible oxidation reactions in the culture medium were also included.

**Table 1. Penetration and percentage of arsenic metabolites after a single exposure to iAs$^{III}$ or MMA$^{III}$ in a rat _in vitro_-BBB model.**

| Arsenic | Time (h) | Penetration rate (%) | | | | | |
|---|---|---|---|---|---|---|---|
| | | TA | iAs$^{III}$ | iAs$^{V}$ | MMA$^{III}$ | MMA$^{V}$ | DMA$^{V}$ |
| iAs$^{III}$ | 0–6 | 12.9±2.23 | 11.2±1.81 | 1.11±0.10 | n.d. | n.d. | 0.90±0.73 |
| | | | (86.9) | (8.72) | | | (4.35) |
| | 0–24 | 43.3±0.71 | 36.6±1.07 | 5.75±0.04 | n.d. | n.d. | 1.42±0.46 |
| | | | (84.6) | (13.3) | | | (2.18) |
| MMA$^{III}$ | 0–6 | 23.1±7.76 | n.d. | n.d. | 7.63±0.79 | 8.27±1.55 | 4.03±0.52 |
| | | | | | (34.6) | (36.9) | (28.5) |
| | 0–24 | 53.1±2.72** | n.d. | n.d. | 13.7±5.09 | 27.9±3.59 | 9.01±0.14 |
| | | | | | (25.9) | (52.7) | (21.4) |

Abbreviations for the five arsenic species: arsenite (iAs$^{III}$); arsenate (iAs$^{V}$); monomethylarsonous acid (MMA$^{III}$); monomethylarsonic acid (MMA$^{V}$); dimethylarsenic acid (DMA$^{V}$). Total arsenic (TA) is the sum of the five arsenic species. The iAs$^{III}$ or MMA$^{III}$ doses added to the rat _in vitro_-BBB model (blood side) were 3,960 ng and 248 ng (as As). Values indicate mean ± SD (n = 3). Values in brackets indicate the percentage of arsenic in total penetration rate. Dimethylarsinous acid was not detected in all samples. Comparison of iAs$^{III}$ and MMA$^{III}$; **$p < 0.01$. n.d., Not detected.

## TJ injury at the BBB following exposure to iAs$^{III}$ and MMA$^{III}$

**Evaluation by TEER and Na-F.** TJ injury at the BBB after a single exposure to iAs$^{III}$ or MMA$^{III}$ was evaluated by %TEER values. These values at 6 h and 24 h showed a statistically significant and concentration-dependent decrease in the iAs$^{III}$ or MMA$^{III}$ groups relative to the control group ($p < 0.001$) (Fig 2). Moreover, treatment with 1μM MMA$^{III}$ reduced TEER significantly compared with 1μM iAs$^{III}$ ($p < 0.001$). Furthermore, this trend was confirmed by treatment with higher concentrations of iAs$^{III}$ (20 μM) or MMA$^{III}$ (2 μM) (Fig 2). Our results show that MMA$^{III}$ produced significantly stronger TJ injury to the BBB than iAs$^{III}$. Next, Na-F values (i.e., the permeability coefficient) were measured using the 24 h specimens that were also used for measuring TEER values. Na-F values after a single treatment with high concentrations of 20 μM iAs$^{III}$ or 2 μM MMA$^{III}$ were then elevated significantly compared to the control group by approximately threefold ($p < 0.05$) for 20 μM iAs$^{III}$ and approximately 16-fold ($p < 0.001$) for 2 μM MMA$^{III}$, respectively (Fig 3). Furthermore, we observed a significant negative correlation between TEER and Na-F values following iAs$^{III}$ or MMA$^{III}$ treatment at 24 h (iAs$^{III}$, $\rho = -0.818$, $p < 0.001$; MMA$^{III}$, $\rho = -0.847$, $p < 0.001$).

## Evaluation using claudin-5 and ZO-1

TJ injury at the BBB following a single treatment with iAs$^{III}$ or MMA$^{III}$ was evaluated by measuring the protein expression patterns of claudin-5 and ZO-1 by WB method. iAs$^{III}$ or MMA$^{III}$ exposure was found to induce a statistically significant concentration-dependent decrease in claudin-5 expression at both 6 h and 24 h compared to the control group ($p < 0.05$, 0.01, 0.001) (Fig 4A and 4B). However, the expression of claudin-5 in 1 μM iAs$^{III}$ and MMA$^{III}$ showed a decreasing pattern in MMA$^{III}$ compared to iAs$^{III}$ at both 6 h and 24 h, but there was no statistically significant difference between the two groups. On the other hand, ZO-1 expression after iAs$^{III}$ or MMA$^{III}$ treatment showed a sporadic decreasing trend (Fig 4A and 4C), but there was no statistically significant concentration-dependent decrease in ZO-1 expression at both 6 h and 24 h compared to the control group. In addition, ZO-1 expression after 20 μM iAs$^{III}$ exposure decreased at 24 h compared to 6 h ($p < 0.05$). Next, the evaluation of the correlation between the expression patterns of claudin-5 and ZO-1 revealed a significant correlation between them in MMA$^{III}$-treated cells (6 h, $\rho = 0.847$, $p < 0.001$) but not in iAs$^{III}$-treated cells (Table 2).

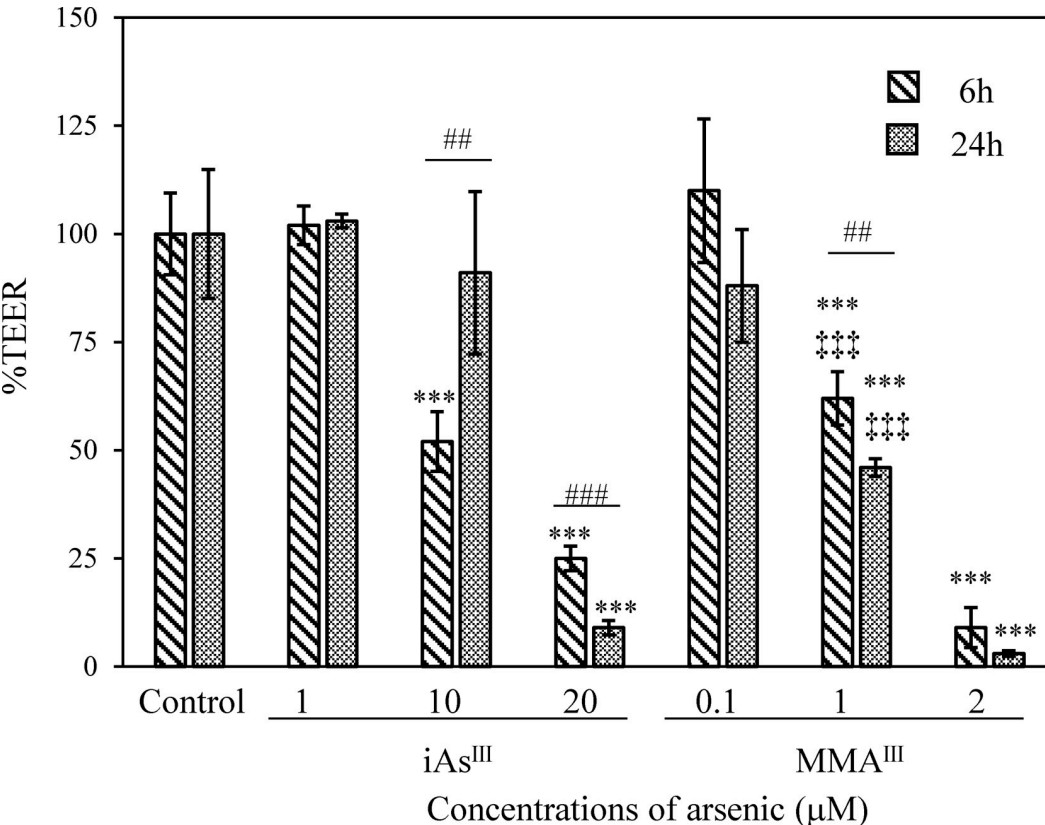

**Fig 2. Changes in TEER following iAs$^{III}$ or MMA$^{III}$ exposure.** TEER was measured at 6 h and 24 h after exposure to iAs$^{III}$ (1, 10, and 20 μM) or MMA$^{III}$ (0.1, 1, and 2 μM). The %TEER was expressed as 100% of the mean of the control TEER value (100%). Results are presented as mean ± SD (n = 4). Comparisons between control and arsenic-treated groups were then performed using one-way ANOVAs with Tukey's post hoc tests at a significance level of ***$p < 0.001$. Student's t-tests were then used to compare TEER values after 6 h and 24 h treatment, and significance levels are indicated as follows: ##$p < 0.01$ and ###$p < 0.001$. Finally, a comparison of the iAs$^{III}$ (1 μM) or MMA$^{III}$ (1 μM) exposure groups was performed by a one-way ANOVA with Tukey's post hoc test, with significance levels indicated as ‡‡‡$p < 0.001$.

Next, TJ injury at the BBB after a single exposure to high concentrations of iAs$^{III}$ (20 μM) or MMA$^{III}$ (2 μM) was evaluated by fluorescent immunostaining for the expression of the claudin-5 and ZO-1 proteins. Claudin-5 expression in the control group was observed as a zonal pattern around the plasma membrane. In the iAs$^{III}$- and MMA$^{III}$- treated groups, a decreased (i.e., sparse and weak) expression of claudin-5 was observed in the regions indicated by red arrows (Fig 5). Claudin-5 expression was decreased at 6 h and 24 h in both the iAs$^{III}$- and MMA$^{III}$-treated groups. In contrast, ZO-1 expression was mildly decreased following treatment with iAs$^{III}$ (20 μM) or MMA$^{III}$ (2 μM) relative to the control group (Fig 5A and 5B). For this phenomenon, quantitative analysis of immunofluorescence staining of claudin-5 and ZO-1 after exposure to iAs$^{III}$ or MMA$^{III}$ was performed (Fig 5C and 5D). Claudin-5 expression was found to be significantly reduced compared to the control ($p < 0.05, 0.01, 0.001$) (Fig 5C), supporting the results of fluorescent immunostaining (Fig 5A). In addition, the decrease in expression tended to be slightly stronger at 24 h than at 6 h (Fig 5C). In contrast, reduction of ZO-1 expression was observed after treatment with iAs$^{III}$ or MMA$^{III}$, as well as claudin-5, compared to the control group (Fig 5B). Quantitative results of ZO-1 expression showed a significant decrease ($p < 0.05, 0.001$) compared to the control group (Fig 5D), supporting the results of fluorescent immunostaining (Fig 5B). In addition, there was a significant decrease in the expression of ZO-1 (Fig 5D), but it tended to be less specific than claudin-5 (Fig 5C).

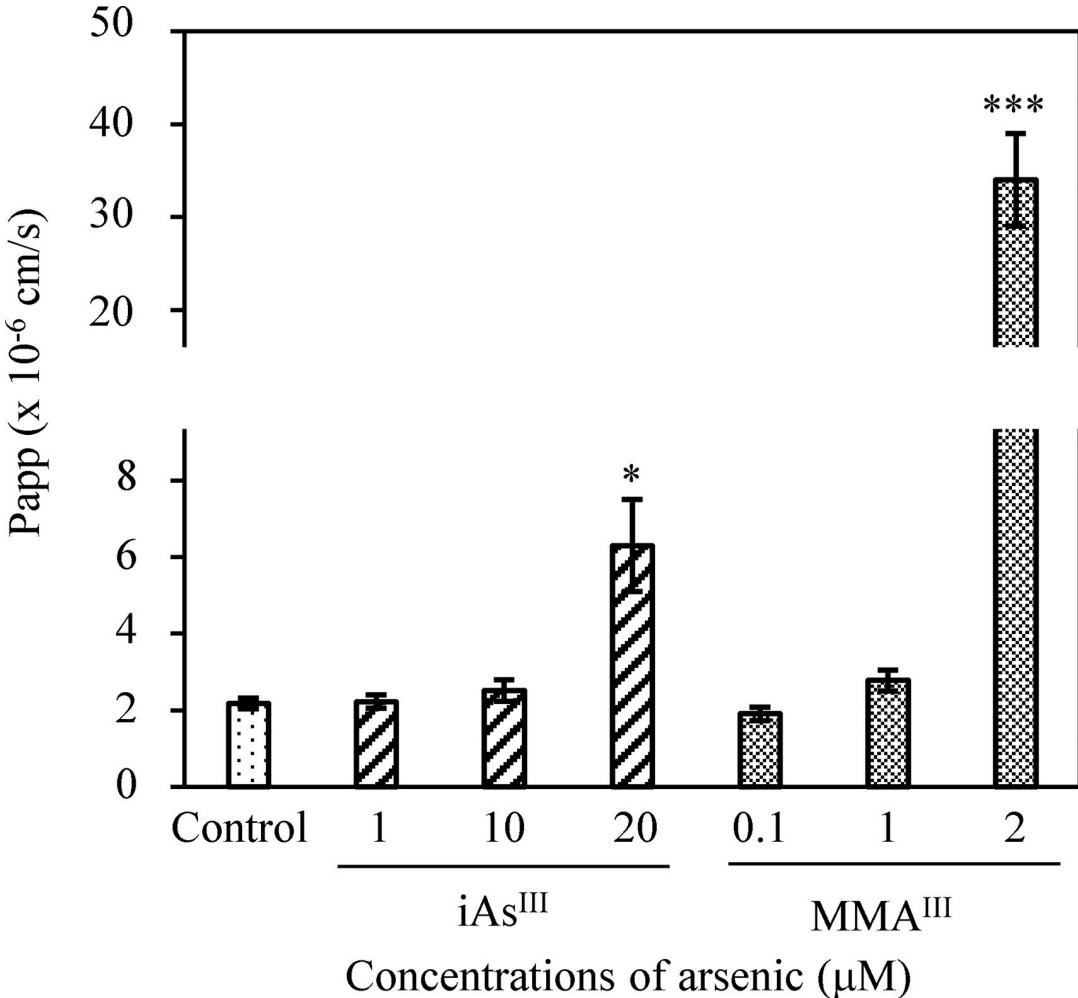

**Fig 3. Evaluation of sodium fluorescein (Na-F) permeability caused by iAs$^{III}$ or MMA$^{III}$ exposure.** iAs$^{III}$ (1, 10, and 20 μM) or MMA$^{III}$ (0.1, 1, and 2 μM) permeability coefficients (Papp) of Na-F were measured after 24 h of treatment. Results are presented as mean ± SD (n = 4). Comparisons between control and arsenic-treated groups were performed using one-way ANOVAs with Dunnett's post hoc tests at significance levels of *p < 0.05; ***p < 0.001.

We understand that TEER value and claudin-5 reductions are significant indicators of TJ injury in the BBB. In this study, we found a significant correlation between TEER and claudin-5 expression (Table 2), with MMA$^{III}$-treated cells showing a significant relationship after 24 h of exposure (24 h, ρ = 0.977, p < 0.001). However, the correlation between TEER and ZO-1 expression was weaker than that between TEER and claudin-5 (Table 2).

### Relationship between the Nrf2 and HO-1 expression and TJ injury

We evaluated the expression of Nrf2 in vascular endothelial cells, pericytes (i.e., the EP group), and astrocytes following treatment with iAs$^{III}$ or MMA$^{III}$ by the WB method. Exposure to low concentrations of iAs$^{III}$ (1 μM) or MMA$^{III}$ (0.1 μM) did not alter Nrf2 expression in the EP group or astrocytes compared with the control (Fig 6A and 6B). The 1μM MMA$^{III}$ treatment showed a significant increase in Nrf2 expression in astrocytes at 6 h compared to the control (p < 0.01), and these values were significantly higher compared to those of the 1μM iAs$^{III}$ treatment (p < 0.05). Moreover, exposure to high concentrations of iAs$^{III}$ (10 and 20 μM) or

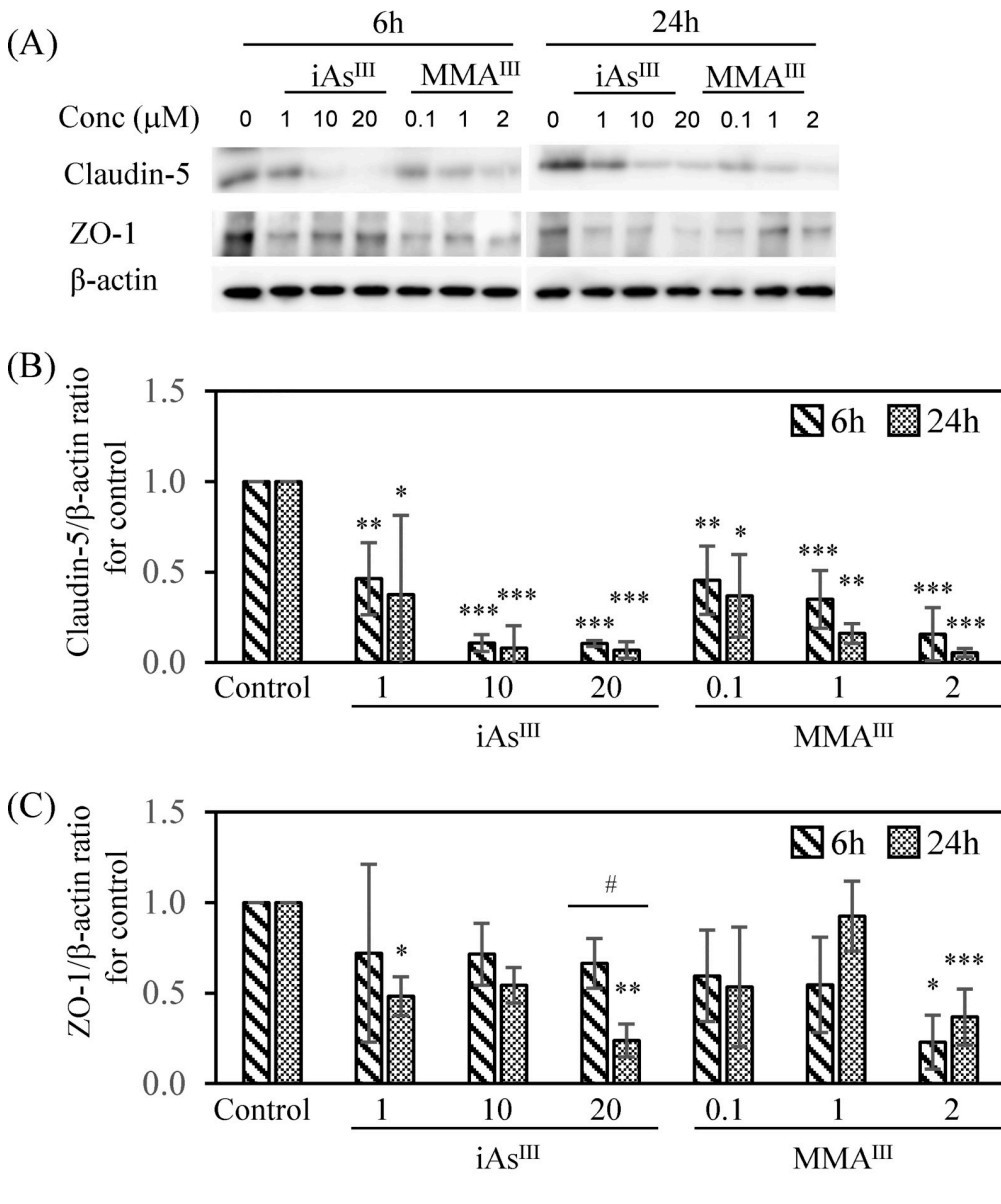

**Fig 4. Expression of the tight junction proteins claudin-5 and ZO-1 following iAs[III] or MMA[III] exposure.** (A) WB analysis of claudin-5 and ZO-1 after 6 h and 24 h of iAs[III] (1, 10, and 20 μM) or MMA[III] (0.1, 1, and 2 μM) exposure. Expression levels of claudin-5 (B) and ZO-1 (C) were quantified by densitometry, standardized to β-actin, and compared with the control value of 1. Results are presented as the mean ± SD of three independent experiments (n = 3). Comparisons between control and arsenic-treated groups were performed using one-way ANOVAs with Tukey's post hoc tests. The significance levels were: $^*p < 0.05$; $^{**}p < 0.01$; and $^{***}p < 0.001$. Finally, differences in the expression of claudin-5 or ZO-1 at 6 and 24 h were tested using Student's t-test. The significance levels were: $^\#p < 0.05$.

MMA[III] (2 μM) was found to increase Nrf2 expression significantly in both the EP group and astrocytes (p < 0.01, 0.001); this effect was particularly significant in astrocytes at the 6 h time point in both the iAs[III]- and MMA[III]-treated groups (Fig 6A and 6B). The expression of HO-1 in the EP group and astrocytes after treatment with iAs[III] or MMA[III] was then evaluated by the WB method. Exposure to iAs[III] increased HO-1 expression significantly in both the EP group

**Table 2. Spearman correlation coefficients (ρ) among TEER, claudin-5, and ZO-1.**

| Sample | Time (h) | Claudin-5 | | TEER | |
|---|---|---|---|---|---|
| | | iAs$^{III}$ | MMA$^{III}$ | iAs$^{III}$ | MMA$^{III}$ |
| **Claudin-5** | 6 | | | 0.923*** | 0.951*** |
| **Claudin-5** | 24 | | | 0.797** | 0.977*** |
| **ZO-1** | 6 | 0.371 | 0.874*** | 0.406 | 0.860*** |
| **ZO-1** | 24 | 0.448 | 0.657* | 0.790** | 0.564 |

The measured values of TEER, claudin-5/β-actin, and ZO-1/β-actin were used to analyze rank correlation coefficients.

*p < 0.05

**p < 0.01

***p < 0.001.

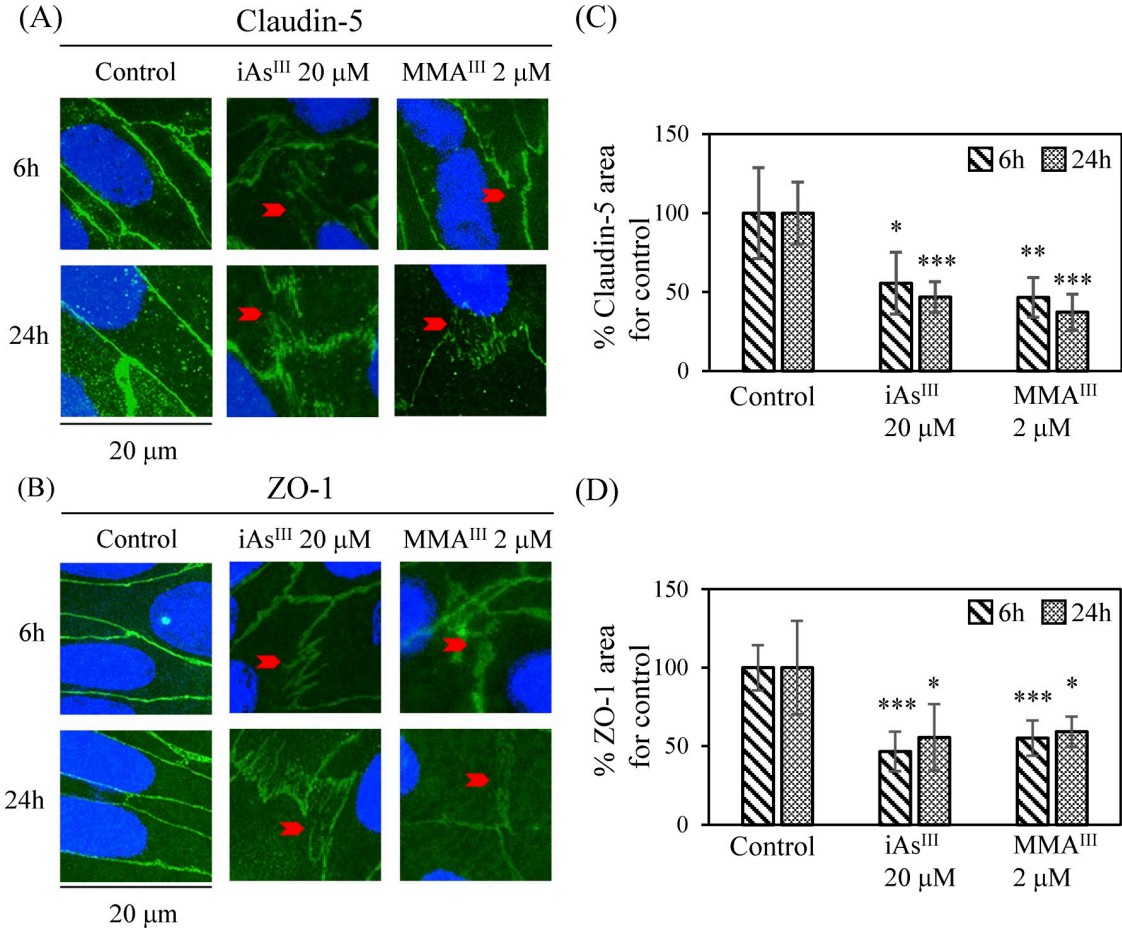

**Fig 5. Altered localization and expression of the tight junction proteins claudin-5 and ZO-1 followed by iAs$^{III}$ or MMA$^{III}$ exposure.** Representative immunofluorescence staining of (A) claudin-5 (green, Alexa Fluor 488) and (B) ZO-1 (green, Alexa Fluor 488), and cell nuclei (blue, DAPI) after 6 h and 24 h exposure to iAs$^{III}$ (20 μM) or MMA$^{III}$ (2 μM). Arrowheads indicate typical localization changes due to tight junction (TJ) damage. Scale bars: 20 μm. Quantitative analysis of immunofluorescence staining of claudin-5 (C) and ZO-1 (D) following iAs$^{III}$ or MMA$^{III}$ exposure. Four other photographs were obtained from the same membrane as those shown in (A) and (B). The % TJ protein expressed area was expressed as 100% of the mean of each control (100%). Results are presented as mean ± SD (n = 5). Comparisons between control and arsenic-treated groups were performed using one-way ANOVAs with Tukey's post hoc tests. The significance levels were: *p < 0.05; **p < 0.01; and ***p < 0.001.

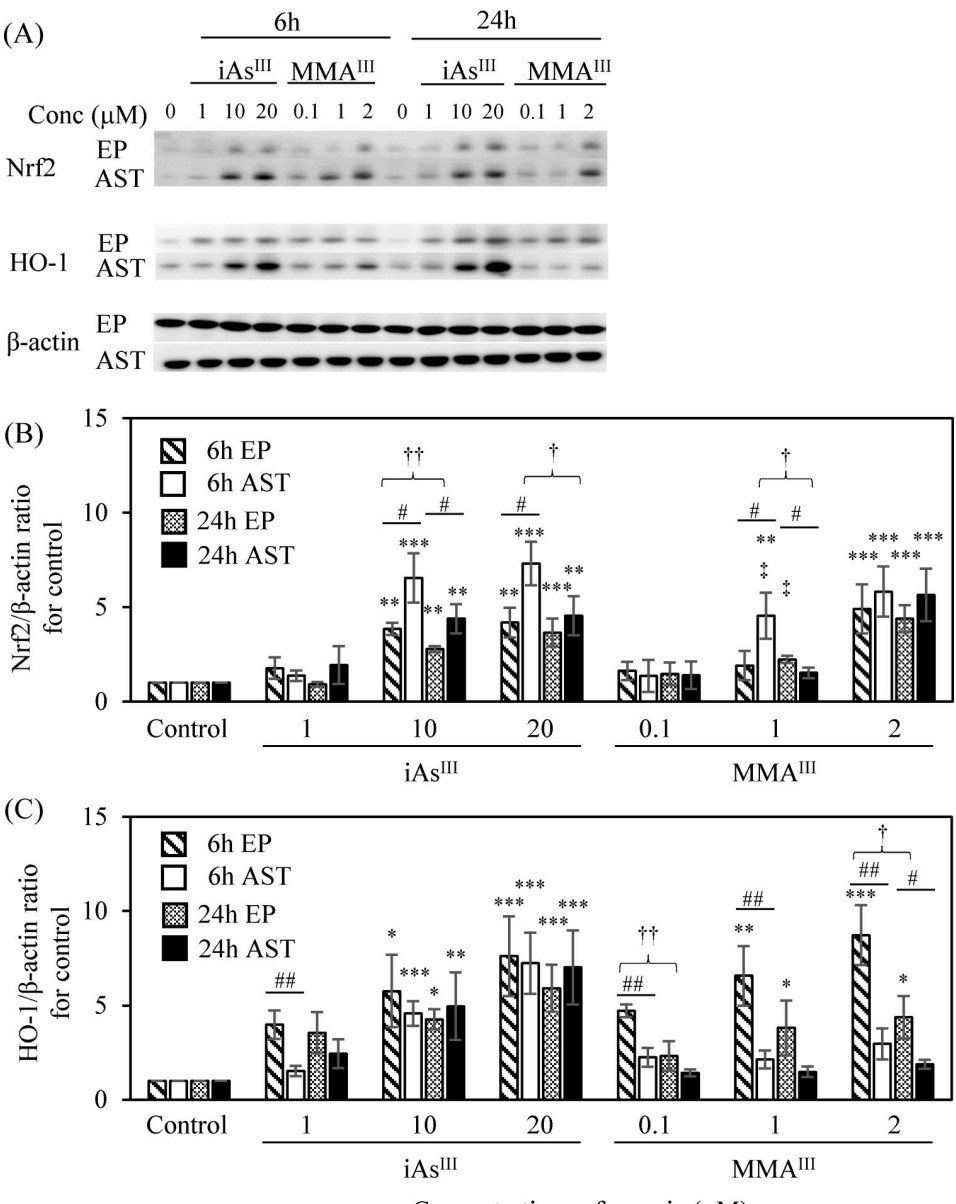

**Fig 6. Expression of Nrf2 and HO-1 in the vascular endothelial cell and pericyte group and astrocytes following iAs$^{III}$ or MMA$^{III}$ exposure.** (A) WB analysis of Nrf2 and HO-1 images after 6 h and 24 h exposure to iAs$^{III}$ (1, 10, and 20 μM) or MMA$^{III}$ (0.1, 1, and 2 μM). EP: vascular endothelial cells and pericyte group; AST: astrocytes. The expression levels of Nrf2 (B) and HO-1 (C) were quantified by densitometry, standardized to β-actin, and compared to the control value of 1. Results are presented as the mean ± SD of three independent experiments (n = 3). Comparison of control and arsenic-treated groups was performed using one-way ANOVAs with Tukey's post hoc tests. The significance levels were: *p < 0.05; **p < 0.01; ***p < 0.001. Moreover, Student's t-tests were used to compare expression between EP and AST, and the significance levels were #p < 0.05 and ##p < 0.01. Student's t-tests were also used to compare 6 h and 24 h treatments, with significance levels of †p < 0.05; ††p < 0.01. Finally, the comparison of iAs$^{III}$ (1 μM) or MMA$^{III}$ (1 μM) exposure groups was performed using one-way ANOVAs with Tukey's post hoc tests, with a significance level of ‡p < 0.05.

and astrocytes except treatment group with 1 μM concentration (Fig 6A and 6C). Exposure to MMA$^{III}$ showed a significant concentration-dependent increase in HO-1 expression in the EP group at 6 h (p < 0.01, 0.001). However, we observed only a slight increase in HO-1 expression

Table 3. Spearman correlation coefficients (ρ) among TEER, Nrf2, and HO-1.

| Sample | Time (h) Cell group | Nrf2 | | TEER | |
|---|---|---|---|---|---|
| | | iAs$^{III}$ | MMA$^{III}$ | iAs$^{III}$ | MMA$^{III}$ |
| **Nrf2** | 6 h EP | | | -0.874*** | -0.657* |
| | 6 h AST | | | -0.874*** | -0.818** |
| | 24 h EP | | | -0.881*** | -0.961*** |
| | 24 h AST | | | -0.811** | -0.835*** |
| **HO-1** | 6 h EP | 0.825*** | 0.825*** | -0.951*** | -0.797** |
| | 6 h AST | 0.874*** | 0.678* | -0.999*** | -0.510 |
| | 24 h EP | 0.818*** | 0.909*** | -0.958*** | -0.947*** |
| | 24 h AST | 0.762** | 0.580* | -0.965*** | -0.821** |

AST, astrocytes; EP, vascular endothelial cell and pericyte group.

*p < 0.05

**p < 0.01

***p < 0.001.

in the astrocytes after 6 h and 24 h (Fig 6A and 6C). Moreover, the increased expression of Nrf2 and HO-1 after iAs$^{III}$ or MMA$^{III}$ exposure is common, so we analyzed the correlation between the two groups (Table 3). Characteristically, the significant correlation between Nrf2 and HO-1 was stronger in the EP group than in astrocytes and more pronounced for iAs$^{III}$ (6 h and 24 h AST, p < 0.01, 0.001) than for MMA$^{III}$ (6 h and 24 h AST, p < 0.05) (Table 3).

Next, we analyzed the correlation between the oxidative stress response and BBB TJ injury after iAs$^{III}$ or MMA$^{III}$ exposure. We have already discussed that TEER values are adequate indicators of BBB TJ injury (Fig 2). In common with iAs$^{III}$ and MMA$^{III}$ exposure, significant negative correlations were identified between Nrf2 and TEER (iAs$^{III}$, p < 0.01, 0.001; MMA$^{III}$, p < 0.05, 0.01, 0.001) or HO-1 and TEER (iAs$^{III}$, p < 0.001; MMA$^{III}$, p < 0.01, 0.001) (Table 3). These results were commonly confirmed in the EP group and astrocytes. In other words, these significant negative correlation coefficients suggest a mechanism by which the anti-stress proteins Nrf2 and HO-1 are induced in a concentration-dependent manner against BBB TJ injury due to oxidative stress caused by iAs$^{III}$ or MMA$^{III}$ exposure. Until now, it has been technically challenging to verify TJ injury in BBB caused by iAs or their metabolites, but the rat *in vitro* BBB model has proven to be an effective and easy method for evaluation.

## Discussion

We hypothesize that the damage to cognitive ability or cognitive dysfunction caused by exposure to iAs results from an initial TJ injury at the BBB, followed by stepwise injury to glial cells and neurons by iAs and their metabolites penetrating the BBB. In this study, a rat *in vitro*-BBB model was used to obtain the first information regarding penetration rate, TJ injury, and oxidative stress responses at the BBB following iAs$^{III}$ and MMA$^{III}$ exposure, as well as arsenic compounds the methylation and oxidative reactions within the BBB that are related to TJ injury.

## Metabolism of iAs$^{III}$ and MMA$^{III}$ in the BBB

We understand that iAs are methylated in the liver and converted to MMA$^{V}$ or DMA$^{V}$ and that some of the arsenic is then transferred from the blood to brain tissue through the BBB [14,15]. In addition, iAs, MMA$^{V}$, and DMA$^{V}$ detected in brain tissue are recognized as a portion of the blood penetrating the BBB. However, studies suggesting a methylation of iAs in

brain tissue are limited. In one experiment, 0.1 μM sodium arsenite (iAs$^{III}$) was added to mouse brain slices, which were then observed for 24–72 h, and DMA$^{V}$ was detected [36]. In another study, 2.5–10 mg/kg/day of sodium arsenite (iAs$^{III}$) was administered orally to mice for nine days, after which their brain tissue was evaluated, revealing the presence of DMA$^{V}$ [37]. Moreover, the methylation of iAs$^{III}$ and MMA$^{III}$ in human astrocytes, microglia, and neurons has been demonstrated; however, to date no methylation has been observed in brain microvascular endothelial cells [33]. We predicted that this methylation capacity exists in primary cultured rat brain cells and used a rat *in vitro*-BBB model to examine the metabolism of iAs$^{III}$ or MMA$^{III}$ in the BBB. The structure of the rat *in vitro*-BBB model (Fig 1) involves a fundamental distinction between the luminal compartment (EP group) and the abluminal compartment (astrocytes region). Different concentrations of iAs$^{III}$ or MMA$^{III}$ were exposed to the luminal compartment and recovered from the abluminal compartment. HPLC-ICP-MS was then used to quantify the arsenic species recovered from the abluminal compartment. This experiment provided information on metabolism within the BBB, and penetration rate was also determined from total metabolite values and exposure.

Examination of samples recovered from the abluminal compartment of the rat *in vitro*-BBB model confirmed that iAs$^{III}$ is methylated and converted to DMA$^{V}$ (Table 1); this is consistent with previous analyses in mice. Furthermore, MMA$^{V}$ was not detected in the abluminal compartment (astrocyte region) in our study. Interestingly, the conversion of iAs$^{III}$ to MMA$^{V}$ has not been confirmed in human astrocytes, microglia, or neurons [33]. Although the mechanism of iAs$^{III}$ methylation in mouse brain tissue is speculative, it has been speculated that AS3MT is involved [36,37]. Although no relevant experiments have been performed in rats, AS3MT is also present in rat brain tissue [38]. Thus, in this study, methylation in the rat *in vitro*-BBB model is assumed to be an effect of AS3MT. Today, it is known that the metabolism of iAs is dependent on arsenic methylation capacity in patients with acute [27] or chronic arsenic poisoning [28–30]; however, the relationship between arsenic methylation capacity and iAs$^{III}$ in human, mouse, or rat brain culture cells has not yet been assessed. Furthermore, validation of MMA$^{III}$ metabolism in mouse brain tissue has yet to be conducted. Interestingly, the results of MMA$^{III}$ methylation in human astrocytes, microglia, and neurons [33] are consistent with the results of the present study (Table 1). Therefore, methylation of iAs$^{III}$ and MMA$^{III}$ in brain tissue is assumed to be a likely mechanism of action in astrocytes. Furthermore, methylation in the rat *in vitro*-BBB model assumes a mechanism of action similar to the one-carbon metabolism [28–30]. We postulate that methylation of iAs$^{III}$ and MMA$^{III}$ in astrocytes may be a protective mechanism for glial cells and neurons. It may represent an independent methylation mechanism in brain tissue that protects against arsenic that has penetrated the BBB.

Next, we note that common oxidation reactions of iAs$^{III}$ or MMA$^{III}$ were observed within the BBB. In the present study, the oxidation reactions of iAs$^{III}$ to iAs$^{V}$ after exposure were low, with values of 8.72% and 13.3% at 6 h and 24 h, respectively. In contrast, the oxidation of MMA$^{III}$ to MMA$^{V}$ after exposure was higher, with values of 36.9% and 52.7% at 6 h and 24 h, respectively. In addition, the oxidation reaction of MMA$^{III}$ to MMA$^{V}$ was confirmed by the culture of human astrocytes, microglia, and neurons, but the conversion of iAs$^{III}$ to iAs$^{V}$ did not yield a clear result [33]. iAs$^{V}$ has a lower acute toxicity than iAs$^{III}$. We believe that the oxidation reaction within the BBB is a detoxification mechanism similar to methylation. It should be noted that this experiment was not conducted under anaerobic conditions, and therefore other oxidation reactions in the culture medium may have affected the results and should therefore be considered.

In summation, approximately 15% of iAs$^{III}$ was detoxified by oxidation and methylation, whereas the remaining 85% was highly toxic. Moreover, approximately 65% of MMA$^{III}$ was detoxified by oxidation and methylation, with the remaining 35% retaining its potent toxicity.

Taken together, we conclude from this information that the TJ injury at the BBB results from the toxic effect of unconverted iAs$^{III}$ (85%) or MMA$^{III}$ (35%).

## TJ injury at the BBB following exposure to iAs$^{III}$ and MMA$^{III}$

Although TEER values are generally used to measure TJ injury in cells, some studies have used TEER values to measure TJ injury in the BBB caused by iAs$^{III}$ exposure. Recently, no change was reported in TEER values in response to exposure to 10 μM iAs$^{III}$ in a porcine *in vitro*-BBB model manufactured from brain capillary endothelial cells [21]. Moreover, in a study using an *in vitro* blood–cerebrospinal fluid barrier model manufactured from porcine choroid plexus epithelial cells, a decrease in TEER values relative to the control was observed in response to 15 μM iAs$^{III}$ exposure [39]. In this study, TEER values following treatment with 1 μM MMA$^{III}$ showed a significant decrease compared with TEER values after 1 μM iAs$^{III}$ treatment ($p < 0.001$). In addition, 2 μM MMA$^{III}$ was found to cause a significant decrease in TEER values compared to 20 μM iAs$^{III}$ (Fig 2) and a corresponding increase in Na-F values (Fig 3). Elevated Na-F levels suggest that the TJs at the BBB are destroyed and iAs$^{III}$ or MMA$^{III}$ penetrates the brain tissue. The rat *in vitro*-BBB model used in this study might have a broader range of sensitivity and utility than the previously reported *in vitro*-BBB models [21] and *in vitro* blood–cerebrospinal fluid barrier model [39].

Next, we investigated the effect of iAs$^{III}$ or MMA$^{III}$ exposure on BBB TJ proteins. In general, the membrane protein claudin-5 accumulates at the base of the gating function of BBB TJs [40], and ZO-1 exists as a backing protein [41]. Therefore, it has been hypothesized that a relationship exists between claudin-5 and ZO-1 expression. In the present study, claudin-5 expression was measured following treatment with 1–20 μM iAs$^{III}$ or 0.1–2 μM MMA$^{III}$ by the WB method. These data showed a concentration-dependent decrease in claudin-5 expression in both groups (Fig 4A and 4B). In addition, using a fluorescent immunostaining method, we observed claudin-5 expression after treatment with 20 μM iAs$^{III}$ or 2 μM MMA$^{III}$. We found tears and defects in the string-like plasma membrane adhesion structure (Fig 5A). In a recent study, oral administration of iAs$^{III}$ to mice 50 mg/L (397 μM) for one month decreased claudin-5 expression in the cerebral cortex [42]. Furthermore, in another study, iAs$^{III}$ (0.15–15 mg/L) was administered orally to pregnant mice, and a decrease in occludin, claudin, ZO-1, and ZO-2 was observed in the brains of their offspring [16,43]. Taken together, these data show similarities between the results of the effect of TJ protein on mouse brain tissue and the results reported here.

This study evaluated the relationship between TJ injury and penetration rate in the BBB. Our TEER and Na-F results, along with the data characterizing claudin-5 and ZO-1 expression, may be evidence of TJ injury in the BBB following exposure to 20 μM iAs$^{III}$ or 2 μM MMA$^{III}$ (Figs 2–5). These results indicate that MMA$^{III}$ treatment expresses TJ injury more intensely than iAs$^{III}$ treatment. In addition, we observed a higher penetration rate (53.1% ± 2.72%) of MMA$^{III}$-treated cells than iAs$^{III}$-treated cells at the 24 h time point (43.3% ± 0.71%; $p < 0.01$; Table 1). To our knowledge, this is the first quantitative finding showing that the penetration rate fluctuates with the degree of TJ injury. However, since this approach is new, we understand the need for further study of rat *in vitro*-BBB models.

## Relationship between the oxidative stress response and TJ injury at the BBB

The activation of Nrf2 is known to inhibit oxidative stress that causes cognitive dysfunction [44], and several studies with anti-stress protein responses to brain neurons by iAs$^{III}$ exposure have been reported. After oral administration of iAs$^{III}$ to mice, activation of Nrf2 and HO-1 in

the cerebral cortex has been observed [43,45], and a single intraperitoneal injection of iAs[III] to mice has also shown activation responses of Nrf2 in the cortex [46]. This study evaluated the impact on Nrf2 and HO-1 expression following TJ injury in the BBB caused by iAs[III] or MMA[III] exposure (Fig 6). After treatment with iAs[III] (10 and 20 μM) or MMA[III] (2 μM), Nrf2 expression was increased significantly in astrocytes. Interestingly, HO-1, a downstream enzyme of Nrf2, was expressed in a concentration-dependent manner in the EP group, which is the site of TJ injury in the BBB. Demonstrating that the antioxidant stress response of Nrf2 and HO-1 plays a vital role in BBB TJ injury is an essential outcome of this study. Finally, we believe that the elimination of oxidative stress must also be thoroughly investigated in studies to elucidate cognitive dysfunction induced by iAs[III] exposure.

## Conclusion

The rat *in vitro*-BBB model can be a potential alternative to *in vivo* studies, especially in the study of metabolism and TJ injury in the BBB, as we have confirmed. Previously, iAs[III] and MMA[III] were found to methylation in human brain cells [33]. In the present rat *in vitro*-BBB model, the methylation and oxidative reaction mechanisms of iAs[III] or MMA[III] were observed, and their conversion to low toxicity MMA[V] and DMA[V] was confirmed, consistent with the results in human brain cells. Interestingly, the TJ injury of BBB was speculated to be the mechanism of unconverted iAs[III] or MMA[III], which maintains substantial toxicity. Furthermore, MMA[III] is more readily metabolized than iAs[III]; however, TJ injury by unconverted MMA[III] has been confirmed to be more potent than iAs[III]. This study provided results that recognize the importance of further examining the toxic effects of MMA[III], a metabolite of iAs. Future research on iAs exposure-induced impairment of cognitive abilities and cognitive dysfunction should be conducted, with research focusing on metabolism functions at the BBB.

## Supporting information

**S1 Table. Changes in TEER following iAs[III] or MMA[III] exposure.** *One-way ANOVAs with Tukey's post hoc tests between control and arsenic-treated groups. Comparison between iAs[III] 1 μM and MMA[III] 1 μM-treated group: p < 0.001 (both 6 h and 24 h treatment). [#]Student's t-tests between 6 h and 24 h treatment.
(XLSX)

**S2 Table. Sodium fluorescein permeability following iAs[III] or MMA[III] exposure.** *One-way ANOVAs with Dunnett's post hoc tests between control and arsenic-treated groups.
(XLSX)

**S3 Table. Expression of claudin-5 and ZO-1 following iAs[III] or MMA[III] exposure.** *One-way ANOVAs with Tukey's post hoc tests between control and arsenic-treated groups. Comparison of expression of claudin-5 between iAs[III] 1 μM and MMA[III] 1 μM-treated group: p = 0.993 and 0.816 (6 h and 24 h treatment); Comparison of expression of ZO-1 between iAs[III] 1 μM and MMA[III] 1 μM-treated group: p = 0.976 and 0.072 (6 h and 24 h treatment). [#]Student's t-tests between 6 h and 24 h treatment.
(XLSX)

**S4 Table. Quantitative analysis of immunofluorescence staining of claudin-5 and ZO-1 following iAs[III] or MMA[III] exposure.** *One-way ANOVAs with Tukey's post hoc tests between control and arsenic-treated groups. [#]Student's t-tests between 6 h and 24 h treatment.
(XLSX)

**S5 Table. Expression of Nrf2 and HO-1 following iAs<sup>III</sup> or MMA<sup>III</sup> exposure.** \*One-way ANOVAs with Tukey's post hoc tests between control and arsenic-treated groups. Comparison of expression of Nrf2 between iAs$^{III}$ 1 μM and MMA$^{III}$ 1 μM-treated group: p = 1.000 and 0.044 (EP group; 6 h and 24 h treatment); p = 0.023 and 0.997 (AST group; 6 h and 24 h treatment). Comparison of expression of HO-1 between iAs$^{III}$ 1 μM and MMA$^{III}$ 1 μM-treated group: p = 0.327 and 1.000 (EP group; 6 h and 24 h treatment); p = 0.952 and 0.910 (AST group; 6 h and 24 h treatment). #Student's t-tests between 6 h and 24 h treatment. †Student's t-tests between EP group and AST group.
(XLSX)

## Acknowledgments

The authors would like to thank Dr. Yang Cao for her technical assistance.

## Author Contributions

**Conceptualization:** Hiroshi Yamauchi.

**Data curation:** Hiroshi Yamauchi, Toshiaki Hitomi, Ayako Takata.

**Formal analysis:** Hiroshi Yamauchi, Toshiaki Hitomi, Ayako Takata.

**Funding acquisition:** Ayako Takata.

**Investigation:** Hiroshi Yamauchi, Toshiaki Hitomi, Ayako Takata.

**Methodology:** Hiroshi Yamauchi, Toshiaki Hitomi.

**Project administration:** Hiroshi Yamauchi.

**Supervision:** Hiroshi Yamauchi, Ayako Takata.

**Validation:** Hiroshi Yamauchi, Toshiaki Hitomi, Ayako Takata.

**Visualization:** Hiroshi Yamauchi, Toshiaki Hitomi, Ayako Takata.

**Writing – original draft:** Hiroshi Yamauchi.

**Writing – review & editing:** Hiroshi Yamauchi, Toshiaki Hitomi, Ayako Takata.

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
