## [Decision Letter · Decision Letter 0]

27 Sep 2023

PONE-D-23-26879PLOS ONE

Evaluation of arsenic metabolism and tight junction injury after exposure to arsenite and monomethylarsonous acid using a rat in vitro blood–brain barrier modelPLOS ONE

Dear Dr. Yamauchi,

Thank you for submitting your manuscript to PLOS ONE. After careful consideration, we feel that it has merit but does not fully meet PLOS ONE’s publication criteria as it currently stands. Therefore, we invite you to submit a revised version of the manuscript that addresses all the points raised during the review process.

Two experts have evaluated the manuscript. It has merit but needs to be improved. While no new experiment is needed, rewriting of the text, and image analysis of the junctional staining, among others are requested.

We look forward to receiving your revised manuscript.

Kind regards,

Mária A. Deli, M.D., Ph.D.

Academic Editor

PLOS ONE

Journal Requirements:

 "This work was supported by JSPS KAKENHI Grant No. JP21H03185.

Ayako Takata." 

"This work was supported by JSPS KAKENHI Grant No. JP21H03185."

"This work was supported by JSPS KAKENHI Grant No. JP21H03185.

Ayako Takata."

Reviewers' comments:

Reviewer's Responses to Questions

**Comments to the Author**

1. Is the manuscript technically sound, and do the data support the conclusions?

Reviewer #1: Yes

Reviewer #2: Yes

2. Has the statistical analysis been performed appropriately and rigorously? 

Reviewer #1: Yes

Reviewer #2: Yes

3. Have the authors made all data underlying the findings in their manuscript fully available?

Reviewer #1: Yes

Reviewer #2: Yes

4. Is the manuscript presented in an intelligible fashion and written in standard English?

Reviewer #1: Yes

Reviewer #2: Yes

5. Review Comments to the Author

Reviewer #1: The manuscript: “Evaluation of arsenic metabolism and tight junction injury after exposure to arsenite and monomethylarsonous acid using a rat in vitro blood–brain barrier model.” Describes the use of an in vitro blood brain barrier model and the impact of two arsenic compounds on the barrier integrity itself and their impact on defined tight junction proteins of the barrier. I really appreciate the model itself as it is a coculture of three highly important cell types of the barrier unit., namely endothelial cells, astrocytes and pericytes. There is on main issue from my side, as the rat is known to metabolise arsenic compounds more effectively, until the dimethylated and pentavalent metabolite DMA(V), that is known to be less toxic compared to for instance the trivalent and monomethylated metabolites. The rat as model to test for arsenic toxicity is not the best suitable one. Please add this into the conclusion. But the overall structure and information of the manuscript is very interesting, the introduction is written clear and leads into the topic, the material and methods section is clear as well. Please add here what constitution the DPBS has and what is the difference between DBPS(+/+) and DBPS. Regarding the sample preparation for the ICP measurements, did you check if you loose analyte after filtration due to binding of the arsenic compounds to the filter. As this is very likely, which filter material has been used? The link for better quality figures leads to wrong figures (e.g. the high quality figure for figure 1 is not available for me, and the one in the manuscript is too blurred). For all figure legends, you only show data +SD not +/-, please correct this. The abbreviation for sodium fluorescein (NaF) is miss-leading, as for chemists it would mean sodium fluoride, please change this. The unit for litre is a big letter (L). Regarding the TEER measurement, at 20 µM iAs(III) and 20 µM MMA(III) the TEER is so long that it can be concluded the barrier is broken, but why is the arsenic quantification in both compartments not the same as there is no barrier apparent anymore? To conclude, after some minor changes, this manuscript should be ready for publication, in my opinion.

Reviewer #2: This manuscript of Yamauchi et al. focuses on an important topic, investigation the tight junction (TJ) impairment of the blood-brain barrier (BBB) caused by arsenic exposure and also the possible metabolic mechanisms behind it. Authors chose proper model for their study, a rat in vitro triple co-culture model of the BBB. The experiments are well designed. However, some part of the manuscript is written poorly which results in confusion. Those parts should be rewritten.

Comments:

1.) The whole “Evaluation using Claudin-5 and ZO-1” paragraph should be rewritten or clarify. Line 296,297.: you claim that there was no difference in claudin-5 expression after treatment with 1 μM iAsIII or 1 μM MMAIII. Before that, it is said that: “iAsIII or MMAIII exposure was found to induce a statistically significant concentration-dependent decrease in Claudin-5 expression…”. If you mean difference between 1 μM iAsIII and MMAIII treatment groups 2-way ANOVA analysis should be used. Also line 299 is confusing. Do you mean treatment groups at 6h time point showed no significant decrease in contrast to claudin-5 expression (Fig. 4B)?

2.) The immunofluorescent staining of TJ proteins should be evaluated by quantification for intensity and morphology.

Minor:

1.) Line 80-81.: “In addition, iAsIII has been reported to HAVE tight junction (TJ) injury…”. Instead of “have” the word “cause” would be a better choice.

2.) Line 126, 129.: For the sake of simplicity, I would write the sample collection from the abluminal compartment for arsenic analysis in one sentence. Sentence in the line 129 is unnecessary.

3.) Figure 1. legend: there are a lot of information (TEER, arsenic exposure) which are not visualized on the figures, hence those are unnecessary. Missing the abbreviation of ZO-1.

4.) Line 148.: “…luminal compartment of the insert portion…” portion is unnecessary.

5.) Line 151.: the sentence is not correct. The concentration of NaF was measured, not the permeability coefficient. Furthermore, please elaborate what is ARVOx4, which wavelength was used.

6.) Table 1.: missing the abbreviations.

7.) Line 236.: dimethylarsinous acid was not mentioned before or explained why it is important.

8.) Figure 4. legend: not the difference in Claudin-5 and ZO-1 expression was tested with Student’s test but the difference in 6h and 24h if I understand correctly. Also “#p” is not indicated in the legend.

9.) In my opinion, using the “i.e.” regarding the concentrations in all the legends is needless.

10.) If claudin-5 is not the beginning of a sentence, it should be written in lowercase.

11.) Figure 6C.: meaning of different columns is missing from the panel. The increase that 1 μM iAsIII treatment caused was not significant (as you say it in line 354-355)?

12.) In Figure 6 related paragraph, please correct the references for the proper figure letter (e.g. line 345.: Fig 6AB; line 354.: Fig 6AC…).

13.) Instead of the word transmittance, “penetration” is more proper.

14.) Below Tables text should be written continuously, not in separate lines.

15.) Line 251.: Instead of (), I would write “Values in brackets indicate…”.

16.) The signs indicating significance (*, #, …) should be positioned above the columns in Figure 2, 3.

17.) Line 275.: The %TEER was expressed as 100% of the mean of control TEER value (100%).

18.) Paragraph from 319-327.: instead of “suppressed” other words should be used, for example decreased.

19.) Line 338.: instead of “owing”, please use “due”.

20.) Line 353, 354.: “…increased HO-1 expression significantly in both the EP group and astrocytes (p < 0.05, 0.01, 0.001)…” I would add to the sentence: except in treatment group with 1 μM concentration.

21.) There is no need to add (p < 0.05, 0.01, 0.001) in the text, only in the legends.

6. PLOS authors have the option to publish the peer review history of their article (what does this mean?). If published, this will include your full peer review and any attached files.

Reviewer #1: No

Reviewer #2: No

---

## [Author Response · Author response to Decision Letter 0]

7 Nov 2023

Confirmed.

We will obey the academic editors and PLOS ONE in all matters.

November 7, 2023

Response to Reviewers

Dear Reviewers,

Thank you for the thoughtful and constructive feedback you provided regarding our manuscript. The following is a point-by-point discussion of our responses to the issues that were raised in the reviews.

Response to Reviewer #1：

Reviewer #1: The manuscript: “Evaluation of arsenic metabolism and tight junction injury after exposure to arsenite and monomethylarsonous acid using a rat in vitro blood–brain barrier model.” Describes the use of an in vitro blood brain barrier model and the impact of two arsenic compounds on the barrier integrity itself and their impact on defined tight junction proteins of the barrier. I really appreciate the model itself as it is a coculture of three highly important cell types of the barrier unit., namely endothelial cells, astrocytes and pericytes. There is on main issue from my side, as the rat is known to metabolise arsenic compounds more effectively, until the dimethylated and pentavalent metabolite DMA(V), that is known to be less toxic compared to for instance the trivalent and monomethylated metabolites. The rat as model to test for arsenic toxicity is not the best suitable one. Please add this into the conclusion. But the overall structure and information of the manuscript is very interesting, the introduction is written clear and leads into the topic, the material and methods section is clear as well. Please add here what constitution the DPBS has and what is the difference between DBPS(+/+) and DBPS. Regarding the sample preparation for the ICP measurements, did you check if you loose analyte after filtration due to binding of the arsenic compounds to the filter. As this is very likely, which filter material has been used? The link for better quality figures leads to wrong figures (e.g. the high quality figure for figure 1 is not available for me, and the one in the manuscript is too blurred). For all figure legends, you only show data +SD not +/-, please correct this. The abbreviation for sodium fluorescein (NaF) is miss-leading, as for chemists it would mean sodium fluoride, please change this. The unit for litre is a big letter (L). Regarding the TEER measurement, at 20 µM iAs(III) and 20 µM MMA(III) the TEER is so long that it can be concluded the barrier is broken, but why is the arsenic quantification in both compartments not the same as there is no barrier apparent anymore? To conclude, after some minor changes, this manuscript should be ready for publication, in my opinion.　

We understood Reviewer #1's comment to be 8 points. We have also described our response to each of the comments below.

Comment 1: The rat as model to test for arsenic toxicity is not the best suitable one. Please add this into the conclusion.

Reviewer 1 comments are important knowledge for arsenic researchers to understand, and we do understand them.

We have extensive experience in arsenic metabolism in patients with acute and chronic arsenic poisoning. On the other hand, we also have extensive experience in studying the metabolism, accumulation, and excretion of inorganic and methylated arsenic compounds in animals (hamsters, mice, and rats). We understand from experience that rats are a special kind of animal, with strong binding of arsenic to erythrocytes, making them unsuitable for urinary excretion studies. However, it is widely understood that rats, like hamsters and mice, have no problems studying arsenic metabolism. Furthermore, our study used primary cultured rat brain cells, but the blood was thoroughly washed away.

Next, we added two papers relevant to the reviewers' comments: Vahter and Marafante et al. report the following observations. In rats, urinary excretion is slower than in mammals such as humans, mice, and hamsters because the metabolically produced DMAV is retained in the erythrocytes, and arsenic is stored in the body for a longer time.

34. Vahter M. Biotransformation of trivalent and pentavalent inorganic arsenic in mice and rats. Environ Res. 1981;25(2): 286-93. https://doi.org/10.1016/0013-9351(81)90030-X 

PMID: 7274192.

35. Marafante E, Bertolero F, Edel J, Pietra R, Sabbioni E. Intracellular interaction and biotransformation of arsenite in rats and rabbits. Sci Total Environ. 1982; 24(1): 27-39. https://doi.org/10.1016/0048-9697(82)90055-9 PMID: 7112092.

The “Revised Manuscript with Track Changes” describes the revised article.

The following sentence was added.

Corrected sentence. Line 119 to 122.: In general, rats are not the best model for arsenic clearance studies because of the strong binding properties of rat erythrocytes to arsenic [34, 35]. In the present study, we used primary cultured rat brain cells washed with intracerebrovascular blood to ensure that rat erythrocytes do not bind arsenic and affect clearance.

Comment 2: Please add here what constitution the DPBS has and what is the difference between DBPS(+/+) and DBPS 

Here is the response to the comment and the revised sentence. DPBS (+/+) is DPBS with additional calcium and magnesium.

Original sentence. Line 186 to 190.: For claudin-5, vascular endothelial cells on transwell inserts were first fixed with a 50% acetone/50% methanol (1:1, ice-cold) solution for one minute at room temperature, and for ZO-1, cells were fixed in 3% paraformaldehyde in DPBS (+/+), DPBS with calcium and magnesium, for 10 min at room temperature then treated with 0.1% Triton X-100 in DPBS (+/+) for 10 min at room temperature.

Corrected sentence. Line 190 to 194.: For claudin-5, vascular endothelial cells on transwell inserts were first fixed with a 50% acetone/50% methanol (1:1, ice-cold) solution for one minute at room temperature, and for ZO-1, cells were fixed in 3% paraformaldehyde in DPBS (+/+), DPBS with calcium and magnesium, for 10 min at room temperature then treated with 0.1% Triton X-100 in DPBS (+/+) for 10 min at room temperature.

Comment 3: Regarding the sample preparation for the ICP measurements, did you check if you loose analyte after filtration due to binding of the arsenic compounds to the filter. As this is very likely, which filter material has been used? 

We have experience with this method of arsenic measurement in a wide variety of samples, including human urine, cell culture medium, and natural water. However, we have not applied this method to samples containing high concentrations of protein. We only use samples that can be diluted with Milli-Q water. The measurement of arsenic in human urine using 0.45 µm membrane filters is commonly found among researchers using the HPLC-ICP- MS method. 

The 0.45 µm membrane filters are made from the device material, polypropylene; the material, hydrophilic polyethersulfone, can be found in the Millipore Filters product catalog.

In this experiment, the adsorption rate of arsenic on 0.45 µm membrane filters has not been verified in detail using five arsenic mixtures and samples collected from the rat in vitro-BBB model. However, we have previously demonstrated the recovery rate using the standard addition method with human urine and confirmed that it is >90%.

　 Original sentence. Line 203 to 204.: Samples were filtered through 0.45 µm membrane filters (Millipore, MA, USA) before analysis.

Corrected sentence. Line 208 to 209.: Samples were filtered through 0.45 µm membrane filters (Hillex-HP, PES 33 mm, Non-sterile, Millipore, MA, USA) before analysis.

Comment 4: The link for better quality figures leads to wrong figures (e.g. the high quality figure for figure 1 is not available for me, and the one in the manuscript is too blurred).

Figures 1 to 6 have changed to high-quality TIFF files.

Comment 5: For all figure legends, you only show data +SD not +/-, please correct this. 

In Figures 2 to 6, ±SD was added to all graph legends.

Comment 6: The abbreviation for sodium fluorescein (NaF) is miss-leading, as for chemists it would mean sodium fluoride, please change this. 

NaF is mistaken for sodium fluoride, so we used sodium fluorescein (Na-F). In the manuscript, we changed to Na-F in 13 places.

Relatedly, we have deleted “sodium” in Line 264.

Original sentence. Line 264 to 266.: Next, sodium NaF values (i.e., the permeability coefficient) were measured using the 24 h specimens that were also used for measuring TEER values.

Corrected sentence. Line 269 to 271.: Next, sodium Na-F values (i.e., the permeability coefficient) were measured using the 24 h specimens that were also used for measuring TEER values.

Comment 7: The unit for litre is a big letter (L). 

The unit of liter was changed to a large letter (L).

Comment 8: Regarding the TEER measurement, at 20 µM iAs(III) and 20 µM MMA(III) the TEER is so long that it can be concluded the barrier is broken, but why is the arsenic quantification in both compartments not the same as there is no barrier apparent anymore? 

It was difficult to understand this reviewer's comment, and we hope we can respond accurately. First of all, the reviewer's comment “20 µM MMA(III)” is mistaken, the correct value is “2 µM MMA(III)”. So, the comment, “but why is the arsenic quantification in both compartments not the same as there is no barrier apparent anymore?” may not be a valid premise. That is exposure of the rat in vitro-BBB model to 20 µM iAs (III) or 2 µM MMA(III) results in different arsenic determination values in both compartments. We hope you will understand our answer.

Response to Reviewer #2：

Reviewer #2. Comments:

1.) The whole “Evaluation using Claudin-5 and ZO-1” paragraph should be rewritten or clarify. Line 296,297.: you claim that there was no difference in claudin-5 expression after treatment with 1 μM iAsIII or 1 μM MMAIII. Before that, it is said that: “iAsIII or MMAIII exposure was found to induce a statistically significant concentration-dependent decrease in Claudin-5 expression…”. If you mean difference between 1 μM iAsIII and MMAIII treatment groups 2-way ANOVA analysis should be used. Also line 299 is confusing. Do you mean treatment groups at 6h time point showed no significant decrease in contrast to claudin-5 expression (Fig. 4B)?

In response to Reviewer 2's comment 1), all relevant text was revised as instructed. The revised contents are shown below.

Original sentence. Line 292 to 302.: TJ injury at the BBB following a single treatment with iAsIII or MMAIII was evaluated by measuring the protein expression patterns of Claudin-5 and ZO-1 by WB method. iAsIII or MMAIII exposure was found to induce a statistically significant concentration-dependent decrease in Claudin-5 expression at both 6 h and 24 h compared to the control group (p < 0.05, 0.01, 0.001) (Fig 4A, B). Interestingly, we found no significant difference in Claudin-5 expression following 1 µM iAsIII or 1 µM MMAIII treatment. On the other hand, ZO-1 expression after iAsIII or MMAIII treatment showed a decreasing trend (Fig 4A, C). However, it was not significant compared to Claudin-5 expression. Next, the evaluation of the correlation between the expression patterns of Claudin-5 and ZO-1 revealed a significant correlation between them in MMAIII-treated cells (6 h, ρ = 0.847, p < 0.001) but not in iAsIII-treated cells (Table 2).

Corrected sentence. Line 297 to 310.: TJ injury at the BBB following a single treatment with iAsIII or MMAIII was evaluated by measuring the protein expression patterns of claudin-5 and ZO-1 by WB method. iAsIII or MMAIII exposure was found to induce a statistically significant concentration-dependent decrease in claudin-5 expression at both 6 h and 24 h compared to the control group (p < 0.05, 0.01, 0.001) (Fig 4A, B). However, the expression of claudin-5 in 1 µM iAsIII and MMAIII showed a decreasing pattern in MMAIII compared to iAsIII at both 6 h and 24 h, but there was no statistically significant difference between the two groups. On the other hand, ZO-1 expression after iAsIII or MMAIII treatment showed a sporadic decreasing trend (Fig 4A, C), but there was no statistically significant concentration-dependent decrease in ZO-1 expression at both 6 h and 24 h compared to the control group. In addition, ZO-1 expression after 20 µM iAsIII exposure decreased at 24 h compared to 6 h (p < 0.05). Next, the evaluation of the correlation between the expression patterns of claudin-5 and ZO-1 revealed a significant correlation between them in MMAIII-treated cells (6 h, ρ = 0.847, p < 0.001) but not in iAsIII-treated cells (Table 2).

2.) The immunofluorescent staining of TJ proteins should be evaluated by quantification for intensity and morphology.　 

In response to reviewer 2 comment 2), we followed the instructions and added new quantitative immunofluorescence staining results for claudin-5 and ZO-1 after iAsIII or MMAIII exposure (Fig. 5C, D). The following sentence was added.

Corrected sentence. Line 335 to 345.: For this phenomenon, quantitative analysis of immunofluorescence staining of claudin-5 and ZO-1 after exposure to iAsIII or MMAIII was performed (Fig 5C, D). Claudin-5 expression was found to be significantly reduced compared to the control (p < 0.05, 0.01, 0.001) (Fig 5C), supporting the results of fluorescent immunostaining (Fig 5A). In addition, the decrease in expression tended to be slightly stronger at 24 h than at 6 h (Fig 5C). In contrast, reduction of ZO-1 expression was observed after treatment with iAsIII or MMAIII, as well as claudin-5, compared to the control group (Fig 5B). Quantitative results of ZO-1 expression showed a significant decrease (p < 0.05, 0.001) compared to the control group (Fig 5D), supporting the results of fluorescent immunostaining (Fig 5B). In addition, there was a significant decrease in the expression of ZO-1 (Fig 5D), but it tended to be less specific than claudin-5 (Fig 5C).

Relatedly, we have added explanations of Figures 5C and 5D to the legend of Figure 5. The following sentence was added.

Corrected sentence. Line 357 to 363.: Quantitative analysis of immunofluorescence staining of claudin-5 (C) and ZO-1 (D) following iAsIII or MMAIII exposure. Four other photographs were obtained from the same membrane as those shown in (A) and (B). The % TJ protein expressed area was expressed as 100% of the mean of each control (100%). Results are presented as mean ± SD (n = 5). Comparisons between control and arsenic-treated groups were performed using one-way ANOVAs with Tukey’s post hoc tests. The significance levels were: *p < 0.05; **p < 0.01; and ***p < 0.001.

We have also added relevant supplementary text in the Immunocytochemistry section of the Methods. The following sentence was added.

Corrected sentence. Line 202 to 203.: Immunofluorescence staining quantification was performed using NIH ImageJ.　 

Minor:

In response to the reviewers' minor comments, we have made deletions, corrections, and additions in 1) through 21).

1.) Line 80-81.: “In addition, iAsIII has been reported to HAVE tight junction (TJ) injury…”. Instead of “have” the word “cause” would be a better choice. 

Corrected sentence. Line 81.: Changed to “cause”.

2.) Line 126, 129.: For the sake of simplicity, I would write the sample collection from the abluminal compartment for arsenic analysis in one sentence. Sentence in the line 129 is unnecessary. 

The original sentence is shown below, with the exception of “Samples for arsenic analysis were collected from the abluminal compartment of the 24-well plates”. The following changes have been made.

　 Original sentence. Line 126 to 130.: and samples taken from the medium in the abluminal compartment (i.e., the brain side) at 6 h and 24 h time points. Samples for Western blot (WB) analysis were collected separately from vascular endothelial cells and pericyte group, or astrocytes. Samples for arsenic analysis were collected from the abluminal compartment of the 24-well plates.

Corrected sentence. Line 129 to 133.: and samples taken from the medium in the abluminal compartment (i.e., the brain side) at 6 h and 24 h time points. Samples for Western blot (WB) analysis were collected separately from vascular endothelial cells and pericyte group, or astrocytes. Samples for arsenic analysis were collected from the abluminal compartment of the 24-well plates.

3.) Figure 1. legend: there are a lot of information (TEER, arsenic exposure) which are not visualized on the figures, hence those are unnecessary. Missing the abbreviation of ZO-1.

Duplicate descriptions of TEER and arsenic concentrations were deleted.

　 Original sentence. Line 135 to 142.: (B) BBB formation was confirmed by measured transendothelial electrical resistance (TEER) values and by immunofluorescence staining the expression of TJ proteins (Claudin-5 and ZO-1; green, Alexa Fluor 488). Mature BBB kits with TEER values greater than 150 Ω × cm2 were used in the experiments. Before arsenic exposure, Claudin-5 and ZO-1 exhibited continuous linear localization (green, Alexa Fluor 488) at the cell membrane (red arrow). Cell nuclei (blue, DAPI) were similarly identified via immunofluorescence staining. iAsIII (i.e., 1, 10, and 20 µM) or MMAIII (i.e., 0.1, 1, and 2 µM) was exposed in the luminal compartment (blood side).

Corrected sentence. Line 138 to 145.: (B) BBB formation was confirmed by measured transendothelial electrical resistance (TEER) values and by immunofluorescence staining the expression of TJ proteins [claudin-5 and zonula occludens-1 (ZO-1); green, Alexa Fluor 488]. Mature BBB kits with TEER values greater than 150 Ω × cm2 were used in the experiments. Before arsenic exposure, Claudin-5 and ZO-1 exhibited continuous linear localization (green, Alexa Fluor 488) at the cell membrane (red arrow). Cell nuclei (blue, DAPI) were similarly identified via immunofluorescence staining. iAsIII (i.e., 1, 10, and 20 µM) or MMAIII (i.e., 0.1, 1, and 2 µM) was exposed in the luminal compartment (blood side).

And the abbreviation of zonula occludens-1 (ZO-1) was corrected.

Corrected sentence. Line 140.: zonula occludens-1 (ZO-1).

4.) Line 148.: “…luminal compartment of the insert portion…” portion is unnecessary.

Corrected sentence. Line 151.: Deleted “portion”.

5.) Line 151.: the sentence is not correct. The concentration of NaF was measured, not the permeability coefficient. Furthermore, please elaborate what is ARVOx4, which wavelength was used. 

Following the instructions, the sentences were changed as follows.

　 Original sentence. Line 150 to 152: After incubation at 37°C for 30 min, the permeability coefficient of NaF was measured using ARVOx4 (Perkin Elmer, Waltham, MA, USA).

Corrected sentence. Line 153 to 156.: After incubation at 37°C for 30 min, the concentration of Na-F was measured using a microplate luminometer ARVOx4 2030 Multilabel Reader (Perkin Elmer, Waltham, MA, USA; excitation wavelength: 485 nm, emission wavelength: 535 nm).

6.) Table 1.: missing the abbreviations.

Abbreviations for the five arsenic species are appended in Table 1. The meaning of “Total” is also added.

Corrected sentence. Line 253 to 255.: Abbreviations for the five arsenic species: arsenite (iAsIII); arsenate (iAsV); monomethylarsonous acid (MMAIII); monomethylarsonic acid (MMAV); dimethylarsenic acid (DMAV). Total is the sum of the five arsenic species. 

7.) Line 236.: dimethylarsinous acid was not mentioned before or explained why it is important.

In response to this point, we have the following opinion: dimethylarsenic acid (DMAV) is known to be the final metabolite in patients with acute and chronic arsenic poisoning caused by inorganic arsenic, and we have provided information on these in the Introduction (lines 93-95). And from metabolism experiments using human brain cells, the final metabolite of monomethylarsonous acid (MMAIII) is also DMAV. This information is given in the Introduction (lines 98-99). Against this background, we did not consider it necessary to include information on DMAV in the Results section (line 241).

8.) Figure 4. legend: not the difference in Claudin-5 and ZO-1 expression was tested with Student’s test but the difference in 6h and 24h if I understand correctly. Also “#p” is not indicated in the legend.

The reviewer 2 comments are correct, and we made a mistake in the notation of the legend in Fig 4. We have corrected it as follows. 

Original sentence. Line 311 to 312.: Finally, the difference in Claudin-5, and ZO-1 expression at 6 h and 24 h was tested using Student’s t-tests.

Corrected sentence. Line 319 to 321.: Finally, differences in the expression of claudin-5 or ZO-1 at 6 and 24 h were tested using Student's t-test. The significance levels were: #p < 0.05.

9.) In my opinion, using the “i.e.” regarding the concentrations in all the legends is needless.

We have removed “i.e.” as noted. Relatedly, we removed “i.e.” regarding the concentrations in the full text.

10.) If claudin-5 is not the beginning of a sentence, it should be written in lowercase. 

We changed the text to “claudin-5” as indicated.

11.) Figure 6C.: meaning of different columns is missing from the panel. The increase that 1 μM iAsIII treatment caused was not significant (as you say it in line 354-355)? 

The reviewer 2 comment is correct, we did not append a legend to Fig 6C. Fig 6C is newly appended with legends for 6h EP, 6h AST, 24h EP, and 24h AST.

Next, we have corrected it as follows.

　 Original sentence. Line 353 to 354.: Exposure to iAsIII increased HO-1 expression significantly in both the EP group and astrocytes (p < 0.05, 0.01, 0.001) (Fig 6).

Corrected sentence. Line 378 to 380.: Exposure to iAsIII increased HO-1 expression

significantly in both the EP group and astrocytes except treatment group with 1 µM

concentration (Fig 6A, C).

Relatedly, the legend in Fig. 6 was also changed.

　 Original sentence. Line 353 to 354.: The levels statistical significance were as follows:

 *p < 0.05; **p < 0.01; ***p < 0.001. Moreover, Student’s t-tests were used to compare expression between EP and AST, and here significance levels were #p < 0.05 and ##p < 0.01.

Corrected sentence. Line 408 to 410.: The significance levels were: *p < 0.05; **p <

 0.01; ***p < 0.001. Moreover, Student’s t-tests were used to compare expression 

 between EP and AST, and the significance levels were #p < 0.05 and ##p < 0.01.

12.) In Figure 6 related paragraph, please correct the references for the proper figure letter (e.g. line 345.: Fig 6AB; line 354.: Fig 6AC…). 

Corrected sentence. Line 367 to 383.: The reviewer's comment is correct, we made a mistake in the notation of Fig 6, correcting it to Fig 6A, B or Fig 6A, C, respectively.

13.) Instead of the word transmittance, “penetration” is more proper.

Corrected sentence Lines 50, 230, 235, 236, 238, 251, 252, 257, 424, 448, 500, 518, 522, 525.: Following the instructions, we changed “transmittance” to “penetration or penetration rate” and applied it to the text and Table 1.

Relatedly, we have changed “migrates into” to “penetrates”.

Original sentence. Line 473 to 474.: Elevated NaF levels suggest that the TJs at the 

BBB are destroyed and iAsIII or MMAIII migrates into the brain tissue.

Corrected sentence. Line 499 to 500.: Elevated Na-F levels suggest that the TJs at the BBB are destroyed and iAsIII or MMAIII penetrates the brain tissue.

14.) Below Tables text should be written continuously, not in separate lines.

Following the instructions, the text below the table was written continuously.

15.) Line 251.: Instead of (), I would write “Values in brackets indicate…”.

The legend in Table 1 has been changed according to the instructions, and the following sentences have also been adjusted.

 Original sentence. Line 251 to 254.: 

() values indicate the percentage of arsenic in total transmittance.

n.d., Not detected. 

dimethylarsinous acid was not detected in all samples.

Comparison of iAsIII and MMAIII; **, p < 0.01.　

Corrected sentence. Line 257 to 259.: Values in brackets indicate the percentage of arsenic in total penetration rate. Dimethylarsinous acid was not detected in all samples. Comparison of iAsIII and MMAIII; **p < 0.01. n.d., Not detected.　

16.) The signs indicating significance (*, #, …) should be positioned above the columns in Figure 2, 3. 

We understood the reviewer 2 comments positively. Because all the significance level signs in Figures 2 and 3 are above the columns. Relatedly, in Figures 4B, C, 5C, D, and 6B, C, the significance level signs are located above the columns as in Figures 2 and 3.

17.) Line 275.: The %TEER was expressed as 100% of the mean of control TEER value (100%).

Followed instructions and corrected.

Corrected sentence. Line 280.: The %TEER was expressed as 100% of the mean of the control TEER value (100%).

18.) Paragraph from 319-327.: instead of “suppressed” other words should be used, for example decreased.　 

Changed “suppressed” to “decreased” in three places in paragraphs 319-327 as instructed.

　 Original sentence. Line 319 to 327.: S Next, TJ injury at the BBB after a single exposure to high concentrations of iAsIII (20 µM) or MMAIII (2 µM) was evaluated by fluorescent immunostaining for the expression of the Claudin-5 and ZO-1 proteins. Claudin-5 expression in the control group was observed as a zonal pattern around the plasma membrane. In the iAsIII- and MMAIII- treated groups, a suppressed (i.e., sparse and weak) expression of Claudin-5 was observed in the regions indicated by red arrows (Fig 5). Claudin-5 expression was suppressed at 6 h and 24 h in both the iAsIII- and MMAIII-treated groups. In contrast, ZO-1 expression was mildly suppressed following treatment with iAsIII (20 µM) or MMAIII (2 µM) relative to the control group (Fig 5).

Corrected sentence. Line 327 to 335.: Next, TJ injury at the BBB after a single exposure to high concentrations of iAsIII (20 µM) or MMAIII (2 µM) was evaluated by fluorescent immunostaining for the expression of the claudin-5 and ZO-1 proteins. Claudin-5 expression in the control group was observed as a zonal pattern around the plasma membrane. In the iAsIII- and MMAIII- treated groups, a decreased (i.e., sparse and weak) expression of claudin-5 was observed in the regions indicated by red arrows (Fig 5). Claudin-5 expression was decreased at 6 h and 24 h in both the iAsIII- and MMAIII-treated groups. In contrast, ZO-1 expression was mildly decreased following treatment with iAsIII (20 µM) or MMAIII (2 µM) relative to the control group (Fig 5A, B).

19.) Line 338.: instead of “owing”, please use “due”.

Corrected sentence. Line 356.: Followed the instructions and changed “owing” to “due”.

20.) Line 353, 354.: “…increased HO-1 expression significantly in both the EP group and astrocytes (p < 0.05, 0.01, 0.001)…” I would add to the sentence: except in treatment group with 1 μM concentration.

21.) There is no need to add (p < 0.05, 0.01, 0.001) in the text, only in the legends.

Follow the instructions and answer 20 and 21) together. 

Original sentence. Line 353 to 354.: Exposure to iAsIII increased HO-1 expression significantly in both the EP group and astrocytes (p < 0.05, 0.01, 0.001) (Fig 6).

Corrected sentence. Line 378 to 380.: Exposure to iAsIII increased HO-1 expression significantly in both the EP group and astrocytes except treatment group with 1 µM concentration (Fig 6A, C).

---

## [Decision Letter · Decision Letter 1]

16 Nov 2023

PONE-D-23-26879; Revised Manuscript, Research article

Evaluation of arsenic metabolism and tight junction injury after exposure to arsenite and monomethylarsonous acid using a rat in vitro blood–brain barrier model

PONE-D-23-26879R1

Dear Dr. Yamauchi,

We’re pleased to inform you that your manuscript has been judged scientifically suitable for publication and will be formally accepted for publication once it meets all outstanding technical requirements.

Kind regards,

Mária A. Deli, M.D., Ph.D.

Academic Editor

PLOS ONE

Additional Editor Comments (optional):

Reviewers' comments:

Reviewer's Responses to Questions

**Comments to the Author**

1. If the authors have adequately addressed your comments raised in a previous round of review and you feel that this manuscript is now acceptable for publication, you may indicate that here to bypass the “Comments to the Author” section, enter your conflict of interest statement in the “Confidential to Editor” section, and submit your "Accept" recommendation.

Reviewer #1: All comments have been addressed

Reviewer #2: All comments have been addressed

2. Is the manuscript technically sound, and do the data support the conclusions?

Reviewer #1: Yes

Reviewer #2: Yes

3. Has the statistical analysis been performed appropriately and rigorously? 

Reviewer #1: Yes

Reviewer #2: Yes

4. Have the authors made all data underlying the findings in their manuscript fully available?

Reviewer #1: Yes

Reviewer #2: Yes

5. Is the manuscript presented in an intelligible fashion and written in standard English?

Reviewer #1: Yes

Reviewer #2: Yes

6. Review Comments to the Author

Reviewer #1: (No Response)

Reviewer #2: (No Response)

7. PLOS authors have the option to publish the peer review history of their article (what does this mean?). If published, this will include your full peer review and any attached files.

Reviewer #1: No

Reviewer #2: No

---

## [Editor Report · Acceptance letter]

20 Nov 2023

PONE-D-23-26879R1 

Evaluation of arsenic metabolism and tight junction injury after exposure to arsenite and monomethylarsonous acid using a rat *in vitro* blood–brain barrier model 

Dear Dr. Yamauchi:

I'm pleased to inform you that your manuscript has been deemed suitable for publication in PLOS ONE. Congratulations! Your manuscript is now with our production department. 

Kind regards, 

on behalf of

Prof. Mária A. Deli 

Academic Editor

PLOS ONE